# Nonparametric Exponential Family Graph Embeddings for Multiple Representation Learning

Chien Lu[1]  Jaakko Peltonen[1]  Timo Nummenmaa[1]  Jyrki Nummenmaa[1]

[1]Tampere University, Finland

## Abstract

In graph data, each node often serves multiple functionalities. However, most graph embedding models assume that each node can only possess one representation. We address this issue by proposing a nonparametric graph embedding model. The model allows each node to learn multiple representations where they are needed to represent the complexity of random walks in the graph. It extends the Exponential family graph embedding model with two nonparametric prior settings, the Dirichlet process and the uniform process. The model combines the ability of Exponential family graph embedding to take the number of occurrences of context nodes into account with nonparametric priors giving it the flexibility to learn more than one latent representation for each node. The learned embeddings outperforms other state of the art approaches in link prediction and node classification tasks.

## 1 INTRODUCTION

Data in the form of graphs is drastically growing across disciplines to represent complex observations and their relationships in the graph topology. One common challenge for such data is unsupervised representation learning (embedding) which discovers underlying functions or characterizations of nodes solely from the graph structure without requiring availability of node attributes. Such research has shown encouragingly that the learned latent representations can be used as features for different predictive tasks with promising performance.

Despite the success of such models, most of the proposed methods consider only the co-appearance pattern of nodes in walks across a graph. The prominence of nodes in their surroundings, for example as hubs or bridges, is an important trait of the network structure but is often ignored.

Moreover, it is a common phenomenon that each graph node can serve different functions or roles: a node can, for example, act both as a local hub for its nearby nodes and also as a crucial bridge along a path between far-off connected areas of a graph. However, most methods are unable to properly represent this: they are restricted to single representation learning where each node is only assigned one latent vector representation. A model that only supports one embedding per node tries to collapse all underlying roles of the node into one vector representation could omit necessary information: this can yield poor representations that are 'inbetween' the roles of the node and do not represent any of them well or represent only some roles while ignoring others.

In this paper we introduce a novel embedding model, which extends exponential family embedding [Rudolph et al., 2016] with nonparametric priors and allows a node to have more than one latent representation. We allocate such latent representations following two nonparametric priors, the Dirichlet process and the uniform process. While Dirichlet processes are popular in nonparametric modeling, the uniform process has been neglected in such models; our results show the uniform process is a promising prior for the proposed model. A tailored truncation-free inference algorithm is developed. Different from the traditional approaches, the algorithm introduces new latent embedding vectors over iterations which provides more efficient inference.

We evaluate the proposed model with two tasks, link prediction and node classification. Results over several datasets show the proposed multiple representation learning method improves performance compared to state of the art baselines.

The contributions of this work are:

- We introduce the notion of multiple representation to graph embeddings: each node can have more than one latent vector representation.

- We propose a graph embedding model leveraging Bayesian nonparametrics, which is unprecedented and challenging to do well. The number of latent represen-

*Accepted for the 38th Conference on Uncertainty in Artificial Intelligence* (UAI 2022).

tations are thus decided by the observed data.

- In addition to the Dirichlet process, we explore the uniform process, and show it is an important option for achieving best results.

- We develop an adaptive inference algorithm for efficient computation.

The paper is organized as follows. Section 2 describes background concepts. Section 3 introduces the proposed model. Section 4 develops the inference algorithm. Experiments are conducted in Section 5 and Section 6 draws the conclusions.

## 2  FUNDAMENTAL CONCEPTS

This section provides a brief overview of some basic concepts that are related to our approach.

### 2.1  EXPONENTIAL FAMILY EMBEDDING

Exponential family embedding (EFE) [Rudolph et al., 2016] is a probabilistic extension of the CBOW embedding model [Mikolov et al., 2013a,b]. Observations are made of objects $v$ that occur at locations $n$ surrounded by a context which is a set of other objects. In a traditional word embedding scenario an object would be a word and the context would be the surrounding words in a sentence; in the graph embedding scenario that we address, objects are instead nodes of a graph and contexts are other nodes on a random walk in the graph.

Let $x_{n,v}$ denote the observed value for object $v$ at location $n$. Denote the context by a set $\mathbf{c}_n = \{v'\}$ of other objects $v'$ and a vector $\tilde{\mathbf{x}}_{\mathbf{c}_n} = \{\tilde{x}_{n,v'}\}$ of their values in the context. In our graph embedding case, the values represent whether the object (graph node) occurs at the location and how many times the context objects (nodes) occur in the context.

In EFE, conditioning on the context set $\mathbf{c}_n$ and context values $\tilde{\mathbf{x}}_{\mathbf{c}_n}$, the observed value $x_{n,v}$ for object $v$ is assumed to be exponential family distributed:

$$x_{n,v}|\mathbf{c}_n, \tilde{\mathbf{x}}_{\mathbf{c}_n} \sim \mathbf{ExpFam}\left(\eta_v\left(\mathbf{c}_n, \tilde{\mathbf{x}}_{\mathbf{c}_n}\right), T\left(x_{n,v}\right)\right) \quad (1)$$

where $\mathbf{ExpFam}$ is an exponential family distribution, $\eta_v\left(\mathbf{c}_n, \tilde{\mathbf{x}}_{\mathbf{c}_n}\right)$ is the natural parameter, and $T\left(x_{n,v}\right)$ denotes the sufficient statistics.

In EFE, each object $v$ is represented in two ways, with an embedding vector $\boldsymbol{\rho}_v \in \mathbb{R}^D$ and a context vector $\boldsymbol{\alpha}_v \in \mathbb{R}^D$ where $D$ is the embedding dimensionality. The EFE captures the co-occurrence pattern by constructing the natural parameter based on interaction between the embedding vector of the center object and the context vectors of its context objects weighted by their context values. The model can be seen as a special generalized linear model since the natural parameter is modeled as a link function of an inner product,

so that

$$\eta_v\left(\mathbf{c}_n, \tilde{\mathbf{x}}_{\mathbf{c}_n}\right) = g\left(\boldsymbol{\rho}_v^\top \frac{1}{|\mathbf{c}_n|} \sum_{v' \in \mathbf{c}_n} \tilde{x}_{n,v'} \boldsymbol{\alpha}_{v'}\right). \quad (2)$$

Since $\mathbf{ExpFam}$ can be any exponential distribution, CBOW can be seen as the special case of employing a Bernoulli distribution where the observed value $x_{n,v}$ can be either 1 or 0. One principal merit of the generalization to other probability distributions is the capability of capturing latent patterns by incorporating the observed values. For example, in a shopping cart scenario, quantity of an observed item is modeled by the quantities of its context items (i.e., other products in the shopping cart) which are not binary but positive integers. Similarly, in a graph embedding scenario counts of graph nodes in a context will be positive integers.

### 2.2  RANDOM WALK BASED NODE EMBEDDING

Let $\mathcal{G} = (\mathbb{V}, \mathbb{E})$ be a graph where $\mathbb{V}$ denotes the set of vertices, and $\mathbb{E} \subseteq \mathbb{V} \times \mathbb{V}$ denotes the edge set. A random walk $\mathbf{w} = \{w_1, \dots, w_L\}$ of length $L$ is a simulated sequence of nodes over the graph where each node is chosen at random from the neighbors of the previous node. Extraction of such random walks is a way to describe a graph by extracting sequence data representing graph connectivity. Such sequences can then be modeled by a generative model.

Random walk based embedding approaches [Perozzi et al., 2014, Grover and Leskovec, 2016] model co-occurrence of nodes in a set of random walks $\mathcal{W}$. The generative process models the sequence content, and thus the graph connectivity, through embeddings of nodes: the model is conditional on the nodes and generates the sequences.

Given a walk $\mathbf{w} \in \mathcal{W}$, the occurrence of node $w_n$ at position $n$ in the walk is conditional on the set $\mathbf{c}_n$ of its surrounding (context) nodes in the walk. The occurrence probability is modeled as depending on embedding vectors of the node and embedding vectors of the context nodes. The representation learning aims to optimize the probability of occurrence of the nodes $w_n$ given their contexts, i.e., $\prod_{\mathbf{w} \in \mathcal{W}} \prod_n p(w_n|\mathbf{c}_n)$.

### 2.3  BAYESIAN NONPARAMETICS

In Bayesian nonparametric models, the number of parameters is not fixed in advance but learned during model fitting up to a potentially infinite number of parameters. The models are typically described as mixtures: each observation is modeled by a parameter drawn from a distribution $G$ over the space of parameters (e.g. $\mathbb{R}^D$) where only a finite number of parameter values have nonzero probability, but $G$ itself is drawn as

$$G \sim NP(G_0, \gamma) \quad (3)$$

from a stochastic process prior $NP$ with base distribution $G_0$ and concentration parameter $\gamma$. The process $NP$ yields distributions over the parameter space, with different numbers of possible values up to a potentially infinite number, but each draw from $NP$ has a finite number. Thus fitting the model to data with the prior $NP$ will infer how many parameters are needed to describe the data.

## 2.4 RELATED WORK

Among random walk based unsupervised node embeddings, Deepwalk [Perozzi et al., 2014] has been the classical method. Grover and Leskovec [2016], Ribeiro et al. [2017] simulate variant random walks emphasizing different structural features of the graph. Celikkanat and Malliaros [2020] extend the models with different likelihoods with EFE framework; in their work, the context vectors are taken to represents the vertices.

A group of models have been proposed to learn multiple representations. Among those, Sun et al. [2019] decide the number of embedding with a community detection task; Liu et al. [2019], Park et al. [2020], Chen et al. [2020] impose a fixed number of embedding vectors for all nodes with a predefined value. The most similar method to ours is Epasto and Perozzi [2019] which uses local neighborhood clustering to generate multiple representations for nodes where different nodes can have different number of embedding vectors. Those methods often depend on extra simulations of the graph data in addition to the random walks data, whereas our method only requires the generated random walks.

Besides random walk based methods, there are other proposed approches include, for example, methods based on matrix factorization [Ou et al., 2016, Wang et al., 2017, Qiu et al., 2018] and neural network based approaches [Li et al., 2018, Velickovic et al., 2019, Wu et al., 2020].

## 3 PROPOSED MODEL

The proposed model is a Bayesian nonparametric extension of exponential family node embedding. We next describe the two notions and how they are used to learn multiple node representations. Figure 1 shows an overall illustration. In the figure, random walks are first extracted from a graph, yielding sequences whose sliding windows each contain a center node and counts of other nodes in the context. The occurrence of the center node will be modeled based on the context, where dependency is characterized using vectorial embeddings: each node has one embedding as a context node and can have multiple embeddings as a center node. The generation of the observed sequence content can be written as a graphical plate representation where nonparametric priors are used to generate the embedding vectors of center nodes, and the center and context embedding vectors together are used to generate observed values, that is, the observed center nodes in each window of a random walk.

### 3.1 EXPONENTIAL FAMILY NODE EMBEDDINGS

Given a simulated random walk node sequence $\mathbf{w} = \{w_1, \ldots, w_L\}$ of length $L$, we slide windows of length $K$ along it. In each window the center node $w_n$ is surrounded by context nodes $\{w_{n-K}, \ldots, w_{n-1}, w_{n+1}, \ldots, w_{n+K}\}$. For each possible vertex $v$ we denote $x_{n,v} = 1$ if it was the center node so that $w_n = v$, otherwise $x_{n,v} = 0$. The context is denoted by the set $\mathbf{c}_n$ of unique vertices in the context nodes and the counts $\tilde{\mathbf{x}}_{\mathbf{c}_n} = \{\tilde{x}_{n,v'}\}$ how many times each vertex $v' \in \mathbf{c}_n$ occurred in them, $\tilde{x}_{n,v'} \leq K - 1$.

We will model dependency of node occurrences along a sequence, based on distributions whose natural parameter compares observed values to their context. In more detail, the natural parameter is based on comparison of node embedding vectors that characterize what kind of surroundings each node tends to appear in. We first describe the distribution and then describe the construction of the natural parameter for different exponential families (different likelihoods).

We model the co-occurrence pattern between $w_n$ and the context $(\mathbf{c}_n, \tilde{\mathbf{x}}_{\mathbf{c}_n})$ with an exponential family

$$x_{n,v} | \mathbf{c}_n, \tilde{\mathbf{x}}_{\mathbf{c}_n} \sim \mathbf{ExpFam}\left(\eta_n\left(\mathbf{c}_n, \tilde{\mathbf{x}}_{\mathbf{c}_n}\right), T\left(x_{n,v}\right)\right) \quad (4)$$

where $\eta_v\left(\mathbf{c}_n, \tilde{\mathbf{x}}_{\mathbf{c}_n}\right)$ is the natural parameter and $T\left(x_{n,v}\right)$ the sufficient statistics.

In this work occurrence of a node is represented as a one-hot choice vector and it is modeled as a draw from an exponential family distribution whose parameters depend on the surrounding nodes. Concretely, if the vertex appears at the location $n$, the positive likelihood is then defined as

$$p(x_{n,v} = 1) = f(x_{n,v} = 1 | \eta_n\left(\mathbf{c}_n, \tilde{\mathbf{x}}_{\mathbf{c}_n}\right), T\left(x_{n,v}\right)) \quad (5)$$

where $f$ is the corresponding probability density function of the exponential family distribution. For a vertex that does not appear at location $n$, the likelihood of the non-appearance (also called a 'negative likelihood') is

$$p(x_{n,v} = 0) = f(x_{n,v} = 0 | \eta_n\left(\mathbf{c}_n, \tilde{\mathbf{x}}_{\mathbf{c}_n}\right), T\left(x_{n,v}\right)) . \quad (6)$$

Since random walks only yield positive samples of vertices that occurred in the center of their windows, learning from them alone would bias the model; thus we use a popular negative sampling approach, and randomly generate several negative samples (5 in experiments) for each location $n$. A negative sample has the same context $(\mathbf{c}_n, \tilde{\mathbf{x}}_{\mathbf{c}_n})$ as the positive sample at $n$, but $x_{n,v}$ is instead set to 1 for a random vertex among those that did not appear in the location. In this work, we explore three different exponential family distributions: Bernoulli, Poisson, and Gaussian.

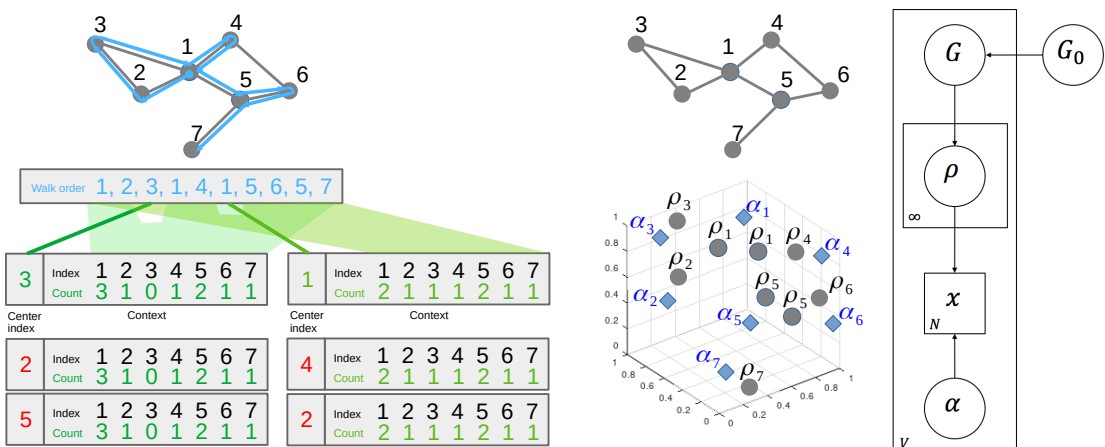

Figure 1: Illustrations of the proposed model. **Left:** random walk (light blue) along a graph from which windows are extracted as positive samples (green) of vertices that were center nodes and counts of other nodes in their context, and corresponding negative samples (red) of vertices that did not occur in the center. **Middle:** each vertex has one or more $d$-dimensional vector representations $\boldsymbol{\rho}$ as center nodes (circles), and one representation $\boldsymbol{\alpha}$ as a context node (diamonds). The picture shows a $d = 3$ dimensional example. **Right:** graphical plate representation of the proposed model.

**Bernoulli Likelihood**. We employ Bernoulli distribution to model the co-occurrence patterns of nodes. Let $\boldsymbol{\rho}_{n,v} \in \mathbb{R}^D$ denote the embedding vector of the node $v$ at the location $n$, $\boldsymbol{\alpha}_v \in \mathbb{R}^D$ denote the embedding vector for the vertex $v$, the natural parameter is then defined as

$$p_n = \mathcal{S}\left(\boldsymbol{\rho}_{n,v}^\top \frac{1}{|\mathbf{c}_n|} \sum_{v' \in \mathbf{c}_n} \boldsymbol{\alpha}_{v'}\right) \qquad (7)$$

where $\mathcal{S}$ denotes the sigmoid function $\mathcal{S} = \frac{1}{1+e^{-x}}$, and $|\mathbf{c}_n|$ is the number of distinct nodes in the context. The appearance of the node $v$ at the location $n$, i.e. whether $x_{n,v} = 1$ or $x_{n,v} = 0$, is thus sampled from a Bernoulli distribution with parameter $p_n$ so that

$$x_{n,v} \sim Bern(p_n) \,. \qquad (8)$$

Note that we use the Bernoulli likelihood to model only the co-appearance of the nodes, which can be seen as an extension of Skip-gram based models. The number of occurrences of nodes in the context is not taken into the account. To incorporate the number of occurrences of nodes, we employ the Poisson and Gaussian distributions.

**Poisson Likelihood**. For a Poisson distribution, the parameter $\lambda_n$ is defined as

$$\lambda_n = \exp\left(\boldsymbol{\rho}_{n,v}^\top \frac{1}{|\mathbf{c}_n|} \sum_{v' \in \mathbf{c}_n} \tilde{x}_{n,v'} \boldsymbol{\alpha}_{v'}\right) \qquad (9)$$

where $|\mathbf{c}_n|$ is again the number of distinct nodes in context and $x_{n,v'}$ denotes the number of occurrences of node $v'$ in the context. The appearance of the node $v$ is generated as

$$x_{n,v} \sim Pois(\lambda_n) \qquad (10)$$

The pivotal difference between the Bernoulli and Poisson cases is that the latter takes the number of occurrences of nodes in the context into account when constructing the natural parameter. The Gaussian case takes the same setting.

**Gaussian Likelihood**. Similar to the settings for Poisson Likelihood, the natural parameter here is defined as

$$\mu_n = \boldsymbol{\rho}_{n,v}^\top \frac{1}{|\mathbf{c}_n|} \sum_{v' \in \mathbf{c}_n} \tilde{x}_{n,v'} \boldsymbol{\alpha}_{v'} \qquad (11)$$

without a specific link function, and the appearance of the node $v$ at the location $n$ is generated as

$$x_{n,v} \sim Norm(\mu_n, \sigma) \qquad (12)$$

where we set $\sigma$ as a fixed hyper-parameter; in the experiments we arbitrarily choose the $\sigma$ from $\{1, 5, 10\}$.

When several different likelihoods are feasible, The model choice can depend on domain expertise, or cross-validation can be used as a model selection process.

### 3.2 NONPARAMETRIC EMBEDDING

Instead of restricting each vertex $v$ to have a single role represented, to better capture the complexity of vertex roles in a graph as observed in random walks, we present a multiple representation learning model which enables each vertex to have multiple latent vector representations, so that the ocurrence of the the vertex at each location in a walk can arise from a different role of the vertex. To do so, we set a nonparametric prior on the embedding vectors $\boldsymbol{\rho}$. That is, we assume that at each location $n$, an embedding vector $\boldsymbol{\rho}_{n,v}$ is generated from a stochastic process $G_v$ specific to

the vertex, so that

$$\boldsymbol{\rho}_{n,v} = \boldsymbol{\rho}_v^{(s)} \sim G_v(G_0, \gamma) \tag{13}$$

where $G_v$ is a stochastic process with a base distribution $G_0$ and a concentration parameter $\gamma$. The base distribution $G_0$ has an infinite number of possible embedding vectors and $G_v$ is a draw from it allocating nonzero probability to a finite number of possibilities $\{\boldsymbol{\rho}_v^{(1)}, \ldots, \boldsymbol{\rho}_v^{(s)}, \ldots, \boldsymbol{\rho}_v^{(S)}, \ldots, \}$ where $S$ is the number of observed embedding vectors. We set the base distribution to be a $d$-dimensional Normal distribution $N(0, \sigma_0)$. In experiments we set $\sigma_0 = 5$ for Bernoulli likelihood and $\sigma_0 = 10$ for both Poisson and Gaussian likelihood. For simplicity, similar to the settings of Rudolph et al. [2017], Rudolph and Blei [2018], although we allow multiple embedding vectors $\boldsymbol{\rho}_{n,v}$ for a vertex we will use only one context vector $\boldsymbol{\alpha}_v$ per vertex; this setting can already generate good results in the experiments, and generalization to allow multiple context vectors is a future work.

In the following, let $\mathbf{n}_v = \mathbf{n}_v^+ \cup \mathbf{n}_v^-$ denote locations related to vertex $v$, so that $\mathbf{n}_v^+$ denotes locations where the $v$ appears and $\mathbf{n}_v^-$ locations where $v$ is the negative sample. Moreover, denote by $\mathbf{n}_{v,<n}$ the subset of $\mathbf{n}_v$ where the location is before $n$, and denote by superscript $(s)$ those locations where the embedding vector was the $s$:th embedding vector of $v$.

**Dirichlet Process**. One of the most common nonparametric process priors is a Dirichlet process. The predictive probability of $\boldsymbol{\rho}_{n,v}$ is defined based on numbers of occurrences of embedding vectors of $v$ at earlier locations $n' < n$ in positive or negative samples, so that

$$P(\boldsymbol{\rho}_{n,v} | \{\boldsymbol{\rho}_{n',v}; n' \in \mathbf{n}_{v,<n}\}) =$$
$$\begin{cases} \frac{|\mathbf{n}_{v,<n}^{(s)}|}{\sum_{s'} |\mathbf{n}_{v,<n}^{(s')}| - 1 + \gamma} & \boldsymbol{\rho}_{v,n} = \boldsymbol{\rho}_v^{(s)}, \forall \boldsymbol{\rho}_v^{(s)} \in \{\boldsymbol{\rho}_v^{(1)} \ldots \boldsymbol{\rho}_v^{(S_v)}\} \\ \frac{\gamma}{\sum_{s'} |\mathbf{n}_{v,<n}^{(s')}| - 1 + \gamma} & \boldsymbol{\rho}_{v,n} = \boldsymbol{\rho}_v^{(S_v+1)} \sim G_0 \end{cases}$$
$$\tag{14}$$

where $|\mathbf{n}_{v,<n}^{(s)}|$ is the number of locations before $n$ where $\boldsymbol{\rho}_v^{(s)}$ has been selected, and $\gamma$ governs the generation of a new embedding vector.

**Uniform process**. An alternative to Dirichlet process is a uniform process [Wallach et al., 2010] with the predictive probability

$$P(\boldsymbol{\rho}_{n,v} | \{\boldsymbol{\rho}_{n',v}; n' \in \mathbf{n}_{v,<n}\}) =$$
$$\begin{cases} \frac{1}{S_v + \gamma} & \boldsymbol{\rho}_{n,v} = \boldsymbol{\rho}_v^{(s)}, \forall \boldsymbol{\rho}_v^{(s)} \in \{\boldsymbol{\rho}_v^{(1)} \ldots \boldsymbol{\rho}_v^{(S_v)}\} \\ \frac{\gamma}{S_v + \gamma} & \boldsymbol{\rho}_{n,v} = \boldsymbol{\rho}_v^{(S_v+1)} \sim G_0 \end{cases}$$
$$\tag{15}$$

where $S_v$ denotes the number of different embedding vectors used for $v$ before location $n$, and the embedding vector $\boldsymbol{\rho}_{n,v}$ is generated independently from the occurrence frequencies of previous generated values. The generation is only controlled by the concentration parameter $\gamma$.

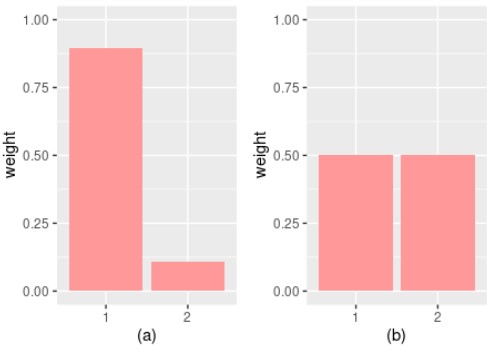

Figure 2: A comparison bewteen two nonparametric priors on the embeedings of the node [YGR078C] in Yeast dataset. (a): Weights of each embedding vector Dp-Pois model ($\gamma = 0.01$). (b): from up-Pois model ($\gamma = 0.000001$).

Despite the popularity of Dirichlet process, it suffers from the "rich get richer" issue, as it tends to repeat previous values and tends to model the first (or first few) embedding vectors as highly dominant, which can limit model flexibility. The uniform process was proposed to address this issue. Figure 2 show an example where the Dirichlet process concentrates on the first embedding vector and uniform process delivers smoother weights. The uniform process has been neglected by the research community, with most applications employing Dirichlet processes as priors.

**Overall generative process.** The proposed model can be summarized with the generative process shown below (corresponding plate model shown in Figure 1, Right):

1. For each vertex $v \in \mathbb{V}$:
    - $G_v \sim NP(G_0, \gamma)$
    - $\boldsymbol{\alpha}_v \sim N(0, \sigma_0^2 I)$

2. For each walk $\mathbf{w} = \{w_1, \ldots, w_L\} \in \mathcal{W}$
    - For location $n$:
        - $\boldsymbol{\rho}_{n,v} \sim G_v$
        - $\eta_{n,v} = g\left(\boldsymbol{\rho}_{n,v}^\top \frac{1}{|\mathbf{c}_n|} \sum_{v' \in \mathbf{c}_n} \tilde{x}_{n,v'} \boldsymbol{\alpha}_{v'}\right)$
        - $x_{n,v} \sim P(\eta_{n,v})$

## 4 INFERENCE

We adapt a truncation-free variational inference algorithm proposed by [Huynh et al., 2016]. Using a stick-breaking construction [Sethuraman, 1994], for vertex $v$ we have

$$G_v = \sum_{s=1}^{\infty} \beta_v^{(s)} \delta_{\boldsymbol{\rho}_v^{(s)}}, \quad \boldsymbol{\rho}_v^{(s)} \sim G_0, \tag{16}$$

$$\beta_v^{(s)} = \zeta_v^{(s)} \prod_{i=1}^{s-1} \left(1 - \zeta_v^{(i)}\right), \quad \zeta_v^{(s)} \sim Beta(1, \gamma). \tag{17}$$

The posterior distribution for the stick breaking parameters $\boldsymbol{\beta}_v = (\beta_v^{(1)}, \ldots, \beta_v^{(S_v)}, \beta_v^{(S_v+1)})$ is then

$$(\beta_v^{(1)}, \ldots, \beta_v^{(S)}, \beta_v^{(S+1)}) \sim Dir(\theta_v^{(1)}, \ldots, \theta_v^{(S_v)}, \gamma) \quad (18)$$

where parameter $\boldsymbol{\theta}_v$ governs the general prevalence over all potential embedding vectors. For each location, the embedding vector $\boldsymbol{\rho}_{n,v}$ is decided by a label $z_{n,v}$ sampled from a Multinomial distribution

$$z_{n,v} \sim Multinomial(\boldsymbol{\beta}_v), \ \boldsymbol{\rho}_{n,v} = \boldsymbol{\rho}_v^{(z_{n,v})}. \quad (19)$$

The variational distribution $q(z_{v,n})$ is updated as

$$\exp\left(E_q\big[\ln z_{n,v}\big]\right) \propto \exp\big(E\big[\ln p(x_{n,v}|\mathbf{c}_n, \tilde{\mathbf{x}}_{\mathbf{c}_n}; \boldsymbol{\rho}_v^{(s)}, \boldsymbol{\alpha})\big] + E\big[\ln p(z_{n,v}|z_{\mathbf{n}_v\backslash n,v}; \gamma)\big]\big) \quad (20)$$

where the first term is the fitness of the selected embedding $\boldsymbol{\rho}_v^{(s)}$, and the second term is related to the prior. If the prior is a Dirichlet process, the second term in Equation (20) is

$$E\left[\ln p(z_{n,v}|z_{\mathbf{n}\backslash n,v}; \gamma)\right] =$$

$$\begin{cases} \ln \dfrac{E[\theta_{\mathbf{n}_v\backslash n,v}^{(s)}]}{|\mathbf{n}_v|-1+\gamma} - \dfrac{1}{2}\dfrac{Var[\theta_{\mathbf{n}_v\backslash n,v}^{(s)}]}{E[\theta_v^{(-s)}]^2} & s \leq S \\ \ln \dfrac{\gamma}{|\mathbf{n}_v|-1+\gamma} & s > S \end{cases} \quad (21)$$

where $\mathbf{n}_v$ denotes the locations of vertex $v$ and $|\mathbf{n}_v|$ denotes its size. We then have

$$E[\theta_v^{(s)}] = \sum_{n \in \mathbf{n}_v} q(z_{n,v} = s) \quad (22)$$

$$E[\theta_{\mathbf{n}_v\backslash n,v}^{(s)}] = \sum_{n \in \mathbf{n}_v\backslash n} q(z_{n,v} = s) \quad (23)$$

$$Var[\theta_v^{(s)}] = \sum_{n \in \mathbf{n}_v} q(z_{n,v} = s)(1 - q(z_{n,v} = s)) \quad (24)$$

$$Var[\theta_{\mathbf{n}_v\backslash n,v}^{(s)}] = \sum_{n \in \mathbf{n}_v\backslash n} q(z_{n,v} = s)(1 - q(z_{n,v} = s)) \quad (25)$$

On the other hand, if the prior is a uniform process, the second term in Equation (20) has a simpler form:

$$E\left[\ln p(z_{n,v}|\gamma)\right] = \begin{cases} \ln \dfrac{1}{|\mathbf{n}_v|+\gamma} & s \leq S \\ \ln \dfrac{\gamma}{|\mathbf{n}_v|+\gamma} & s > S \end{cases} \quad (26)$$

The truncation-free algorithm starts with setting $S = 1$, where $q(z_{v,n}^{(S+1)}) = 0$. When $E[\theta_v^{(S+1)}] > 1$, the algorithm sets $S = S + 1$, increasing the dimension of vector $z_{v,n}$, and sets $q(z_{v,n}^{(S+1)}) = 0$. We can then use the $\theta_v$ to calculate the expected weighting of the vector $\boldsymbol{\rho}_v^{(s)}$.

$$\hat{\beta}_v^{(s)} = E_q\left[\beta_v^{(s)}\right] = \frac{E_q\left[\theta_v^{(s)}\right]}{\sum_{s=1}^{S_v} E_q\left[\theta_v^{(s)}\right]} \quad (27)$$

---

**Algorithm 1:** Inference Algorithm

---

**input** : Random walks $\mathcal{W}$, negative samples $\tilde{\mathcal{W}}$, initial learning rate $\xi$, number of epochs, number of mini-batches $M$

**output** : embedding vectors $\Phi = \{\boldsymbol{\rho}, \boldsymbol{\alpha}\}$, embedding weights $\{\hat{\boldsymbol{\beta}}\}$

**foreach** $v \in \mathbb{V}$ **do**
  *Set $S_v = 1$, initialize embedding vectors $\rho_v^{(1)}$, $\alpha_v$*
**end**
**foreach** *epoch* **do**
  *Divide input data into $M$ random partitions.*
  **for** $m \leftarrow 1$ **to** $M$ **do**
   *Use the subset $\mathcal{W}^{(m)}$ and $\tilde{\mathcal{W}}^{(m)}$*
   **foreach** $v$ **do**
    **foreach** $n \in \boldsymbol{n}_v^{(m)}$ **do**
     *update $z_{n,v}$ with Equation (20)*
    **end**
    *updata $\theta_v$ with Equation (22) - (25)*
    *Calculate $\hat{\beta}_v$ with Equation (27)*
    **if** $E[\theta_v^{(S+1)}] > 1$ **then**
     $S_v = S_v + 1$
     **foreach** $n \in \boldsymbol{n}_v$ **do**
      *increase the dimension of $z_{n,v}$ and set $z_{n,v}^{(S+1)} = 0$*
     **end**
    **end**
   **end**
  **end**
  *update embedding vectors $\Phi = \{\boldsymbol{\rho}, \boldsymbol{\alpha}\}$*
  $\Phi = \Phi - \xi * \frac{\partial \mathcal{L}}{\partial \Phi}$
  $\xi$ *is set with Adam[Kingma and Ba, 2015]*
**end**

---

**Inference of embedding vectors.**

After updating the $E_q\left[z_{n,v}\right]$, the inference is conducted by optimizing the objective function $\mathcal{L} = \mathcal{L}_{prior} + \mathcal{L}_{likelihood}$.

The term $\mathcal{L}_{prior} = \log p(\rho) + \log p(\alpha)$ is derived from the Gaussian prior $N(0, \sigma_0^2)$ for the embedding vectors:

$$\log p(\boldsymbol{\rho}_v^{(s)}) = \frac{\left\|\boldsymbol{\rho}_v^{(s)}\right\|^2}{-2\sigma_0^2}, \ \log p(\boldsymbol{\alpha}_v) = \frac{\left\|\boldsymbol{\alpha}_v\right\|^2}{-2\sigma_0^2}. \quad (28)$$

Table 1: Datasets for Link Prediction

| Data | $\|V\|$ | $\|E\|$ | Avg.deg | Density |
|------|------|--------|---------|---------|
| GitHub | 37700 | 289003 | 15.332 | 0.00041 |
| Wikipedia | 11631 | 180020 | 30.955 | 0.00266 |
| Twitch | 7126 | 35324 | 9.914 | 0.00140 |

Table 2: Datasets for Node Classification

| Data | $\|V\|$ | $\|E\|$ | $\|K\|$ | Avg.deg | Density |
|------|------|--------|------|---------|---------|
| LastFM | 7624 | 27806 | 18 | 7.294 | 0.00095 |
| CiteSeer | 3327 | 4237 | 6 | 2.845 | 0.00043 |
| Yeast | 2617 | 11855 | 13 | 9.060 | 0.00346 |

For Bernoulli likelihood we have

$$\mathcal{L}_{likelihood} = \sum_{v \in \mathbb{V}} ( \sum_{n \in \mathbf{n}_v^+} \sum_{s \in S_v} E_q \left[ z_{n,v} = s \right] p_n +$$
$$\sum_{n \in \mathbf{n}_v^-} \sum_{s \in S_v} E_q \left[ z_{n,v} = s \right] (1 - p_n)) . \quad (29)$$

For Poisson likelihood we have

$$\mathcal{L}_{likelihood} = \sum_{v \in \mathbb{V}} ( \sum_{n \in \mathbf{n}_v^+} \sum_{s \in S_v} E_q \left[ z_{n,v} = s \right] (\log \lambda_n - \lambda_n)$$
$$- \sum_{n \in \mathbf{n}_v^-} \sum_{s \in S_v} E_q \left[ z_{n,v} = s \right] \lambda_n) . \quad (30)$$

For Gaussian likelihood, we have

$$\mathcal{L}_{likelihood} = \sum_{v \in \mathbb{V}} ( \sum_{n \in \mathbf{n}_v^+} \sum_{s \in S_v} E_q \left[ z_{n,v} = s \right] \left( \frac{(1 - \mu_n)^2}{-2\sigma^2} \right)$$
$$+ \sum_{n \in \mathbf{n}_v^-} \sum_{s \in S_v} E_q \left[ z_{n,v} = s \right] \left( \frac{\mu_n^2}{-2\sigma^2} \right)) . \quad (31)$$

We then use gradient descent to update the embedding vectors over iterations.

### 4.1 STOCHASTIC INFERENCE

We employ stochastic inference. For each epoch, the input data is randomly partitioned into $M$ mini-batches and only one mini-batch is used for each iteration. When mini-batch $m$ is used, the sum over locations $\mathbf{n}_v$ can be approximated by a sum over a subsampled set $\mathbf{n}_v^{(m)}$, so the right-hand side of (22) is approximated by $\frac{|\mathbf{n}_v|}{|\mathbf{n}_v^{(m)}|} \sum_{n \in \mathbf{n}_v^{(m)}} q(z_{n,v} = s)$ and similarly in the other sums. The inference procedure is summarized in Algorithm 1. For all the experiments conducted in this work, we run two epochs with 1000 mini-batches and initial learning rate $\xi = 0.01$. For the negative samples, we

generate $\tilde{\mathcal{W}}$ with 5 negative samples for each location following the procedure of Mikolov et al. [2013b], Celikkanat and Malliaros [2020].

## 5 EXPERIMENTS

For generality, we run experiments with two standard tasks commonly adopted in graph embedding works, link prediction and node classification, with 3 data sets [Csardi and Nepusz, 2006, Rossi and Ahmed, 2015, Rozemberczki et al., 2020] for each task (Tables 1 and 2). The data sets cover varied domains and aim to represent typical use scenarios of the proposed method. We denote our method variants by prior (dp: Dirichlet process, up: uniform process) and Exp-Fam distribution (Bern, Pois, Norm), e.g. 'up-emb (Bern)'. We compare to random walk based methods DeepWalk [Perozzi et al., 2014], node2Vec [Grover and Leskovec, 2016], struc2vec [Ribeiro et al., 2017], and EFGE [Celikkanat and Malliaros, 2020], and Splitter [Epasto and Perozzi, 2019]. To evaluate effect of embedding dimensionality, for each method we run three dimension settings: $D = 50, 100$, and $150$. The concentration parameter for our model is chosen from $\gamma = \{0.01, 0.05, 0.1\}$ for Dirichlet process and $\gamma = \{0.0000001, 0.0000005, 0.000001\}$ for uniform process. The input random walks are generated with the R package igraph [Csardi and Nepusz, 2006] with 80 walks per node with length $L = 10$, the random walks are also fed to EFGE. For other methods, parameters are all set to default values.

### 5.1 TASK: LINK PREDICTION

In link prediction, for each graph we first randomly move 50% of the edges into a held-out test set while keeping the remaining training graph connected. In both training and test sets, randomly sampled negative edges are added in equal amount to the positive edges. A classifier is trained based on the reduced training graph and the training negative edges; the classifier is used to classify the held-out test-set edges. As in the previous single-representation learning works including Deepwalk, node2vec, struc2vec, and EFGE, logistic regression is selected as the classifier. In our approach, to incorporate multiple representations when training the classifier, we employ logistic regression with sample weights, embedding $\boldsymbol{\rho}_v^{(s)}$ is weighted by $\hat{\beta}_v^{(s)}$. The Splitter used maximum dot-product similarity, we transform the similarity into a class probability using logistic regression.

Note that when logistic regression is trained with sample weighting, embeddings of all nodes in our model are separate samples weighted in the log-likelihood by their occurrence probabilities. The regression learns to classify nodes based on all their embedding vectors, and at test time, a node is classified by weighted average of class probabilities predicted for each of its embedding vectors. Thus, the

Table 3: Results for Link Prediction

| | GitHub | | | Wikipedia | | | Twitch | | |
| | D = 50 | D = 100 | D = 150 | D = 50 | D = 100 | D = 150 | D = 50 | D = 100 | D = 150 |
|---|---|---|---|---|---|---|---|---|---|
| Deepwalk | 0.722 | 0.695 | 0.694 | 0.911 | 0.915 | 0.922 | 0.659 | 0.649 | 0.672 |
| node2vec | 0.731 | 0.734 | 0.731 | 0.913 | 0.931 | 0.941 | 0.681 | 0.691 | 0.698 |
| struc2vec | 0.849 | 0.864 | 0.874 | 0.820 | 0.881 | 0.863 | 0.830 | 0.828 | 0.840 |
| EFGE (Bern) | 0.729 | 0.726 | 0.736 | 0.939 | 0.950 | 0.962 | 0.681 | 0.687 | 0.707 |
| EFGE (Pois) | 0.728 | 0.771 | 0.771 | 0.950 | 0.955 | 0.964 | 0.679 | 0.708 | 0.714 |
| EFGE (Norm) | 0.862 | 0.868 | 0.888 | 0.977 | 0.983 | 0.985 | 0.791 | 0.791 | 0.802 |
| Splitter | 0.898 | 0.600 | 0.900 | 0.876 | 0.880 | 0.884 | 0.836 | 0.823 | 0.823 |
| dp-emb (Bern) | 0.823 | 0.831 | 0.830 | 0.986 | 0.991 | 0.991 | 0.757 | 0.787 | 0.782 |
| dp-emb (Pois) | 0.737 | 0.723 | 0.780 | 0.979 | 0.984 | 0.986 | 0.656 | 0.704 | 0.716 |
| dp-emb (Norm) | 0.923 | **0.932** | 0.929 | 0.985 | 0.985 | 0.985 | 0.847 | 0.845 | **0.871** |
| up-emb (Bern) | 0.813 | 0.838 | 0.843 | **0.989** | **0.991** | **0.992** | 0.750 | 0.788 | 0.784 |
| up-emb (Pois) | 0.741 | 0.767 | 0.780 | 0.979 | 0.982 | 0.986 | 0.658 | 0.706 | 0.714 |
| up-emb (Norm) | **0.926** | 0.932 | **0.931** | 0.985 | 0.985 | 0.986 | **0.849** | **0.846** | 0.869 |

multiple embedding vectors are treated separately instead of being combined in a simplistic weighted average.

Three different datasets are used for the link prediction task.

**GitHub**: a social network where each node is a GitHub developer, links between nodes are mutual follow relations.
**Wikipedia**: a network of English Wikipedia pages. Edges between pages reflect their mutual links.
**Twitch**: a user-user interaction network between gamers. Edge between two nodes represents mutual friendship.

We evaluate the binary link classification by area under the curve (AUC). Table 3 shows our model performs well on all datasets; the model with Gaussian likelihood works best.

### 5.2 TASK: NODE CLASSIFICATION

In this task, each node has a class. The learned embedding vectors are used as input features to train a classifier to predict the class of each node. Again, for Deepwalk, node2vec, struc2vec and EFGE, a logistic regression classifier is used. For our model, the logistic regression with sample weights is used. For Splitter, we take the same procedure with each embedding equally weighted. Three different datasets are used for the node classification task.

**LastFM Asia**: a network of people living in Asia using the streaming site LastFM. Links represent followership relations. The class of each node is its location.
**CiteSeer**: a scientific publication network from the CiteSeer digital library. Each node belongs to 1 of 6 categories.
**Yeast**: a protein-protein interaction network. The "Class" attribute of each protein is based on its function (e.g. energy).

We evaluate the performance by Micro-averaged F1, reported in Table 4. Our model outperforms other methods. Rozemberczki et al. [2020] Additionaly, in general, our

model took 2-4 hours to converge (depends on different tasks and settings) without GPU. The Splitter, which also learns multiple representations for each node, took 10+ hours on a GPU machine and 100+ hours without GPU. Our approach achieved better results with less resources.

## 6 DISCUSSIONS AND CONCLUSIONS

We proposed nonparametric exponential family graph embedding, allowing multiple node representations, drawn both with a Dirichlet process prior, and also exploring uniform processes. A tailored algorithm for efficient computation is provided. The experiments demonstrate the learned multiple representations can enhance performance in two tasks. We considered three classical exponential family distributions, Bernoulli, Poisson, and Gaussian, which yielded promising results. Our model can be adapted to other distributions such as Geometric and Chi-square with the proposed nonparametric framework. In our experiments, the hyperparameter $\gamma$ of the nonparametric prior was fixed for the nodes, which already yielded promising results in the standard tasks; having differing $\gamma$ values could be useful for extending the model to scenarios such as learning multiple representations for under-represented nodes, or imbalanced classification tasks.

### Acknowledgements

This work is supported by the Academy of Finland, decisions 312395 and 327352.

### References

Abdulkadir Celikkanat and Fragkiskos D Malliaros. Exponential family graph embeddings. In *Proceedings of the*

Table 4: Results for Node Classification

| LastFM | (D = 50) | | | | (D = 100) | | | | (D = 150) | | | |
|---|---|---|---|---|---|---|---|---|---|---|---|---|
| | 10% | 30% | 60% | 90% | 10% | 30% | 60% | 90% | 10% | 30% | 60% | 90% |
| Deepwalk | 0.756 | 0.800 | 0.819 | 0.823 | 0.754 | 0.796 | 0.819 | 0.829 | 0.750 | 0.797 | 0.819 | 0.826 |
| node2vec | 0.741 | 0.796 | 0.820 | 0.828 | 0.741 | 0.802 | 0.824 | 0.829 | 0.740 | 0.799 | 0.826 | 0.834 |
| struc2vec | 0.116 | 0.127 | 0.130 | 0.138 | 0.128 | 0.149 | 0.165 | 0.174 | 0.131 | 0.159 | 0.178 | 0.189 |
| EFGE-Bern | 0.749 | 0.805 | 0.826 | 0.831 | 0.758 | 0.805 | 0.824 | 0.830 | 0.758 | 0.803 | 0.826 | 0.832 |
| EFGE-Pois | 0.741 | 0.791 | 0.820 | 0.825 | 0.743 | 0.793 | 0.817 | 0.822 | 0.745 | 0.798 | 0.821 | 0.825 |
| EFGE-Norm | 0.758 | 0.807 | 0.826 | 0.832 | 0.752 | 0.804 | 0.824 | 0.830 | 0.755 | 0.808 | 0.827 | 0.833 |
| Splitter | 0.428 | 0.519 | 0.541 | 0.546 | 0.426 | 0.490 | 0.530 | 0.573 | 0.451 | 0.469 | 0.533 | 0.567 |
| Dp-Bern | **0.809** | **0.833** | **0.839** | 0.833 | **0.810** | **0.835** | **0.846** | 0.849 | 0.800 | **0.835** | 0.843 | **0.850** |
| Dp-Pois | 0.776 | 0.821 | 0.831 | 0.833 | 0.782 | 0.822 | 0.831 | 0.830 | 0.782 | 0.823 | 0.832 | 0.833 |
| Dp-Norm | 0.751 | 0.807 | 0.822 | 0.823 | 0.740 | 0.804 | 0.820 | 0.821 | 0.744 | 0.807 | 0.823 | 0.831 |
| up-Bern | 0.806 | 0.831 | 0.835 | **0.841** | 0.802 | 0.835 | 0.841 | **0.852** | 0.804 | 0.833 | **0.844** | 0.844 |
| up-Pois | 0.781 | 0.818 | 0.828 | 0.829 | 0.802 | 0.835 | 0.841 | 0.852 | 0.779 | 0.821 | 0.830 | 0.834 |
| up-Norm | 0.754 | 0.811 | 0.822 | 0.823 | 0.742 | 0.805 | 0.821 | 0.823 | 0.733 | 0.806 | 0.821 | 0.827 |

| Citeseer | (D = 50) | | | | (D = 100) | | | | (D = 150) | | | |
|---|---|---|---|---|---|---|---|---|---|---|---|---|
| Deepwalk | 0.432 | 0.479 | 0.487 | 0.519 | 0.453 | 0.497 | 0.520 | 0.530 | 0.459 | 0.504 | 0.525 | 0.532 |
| node2vec | 0.456 | 0.503 | 0.508 | 0.555 | 0.493 | 0.529 | 0.539 | 0.544 | 0.501 | 0.538 | 0.570 | 0.582 |
| struc2vec | 0.224 | 0.240 | 0.278 | 0.314 | 0.226 | 0.250 | 0.274 | 0.294 | 0.224 | 0.243 | 0.254 | 0.297 |
| EFGE-Bern | 0.468 | 0.502 | 0.508 | 0.518 | 0.477 | 0.503 | 0.516 | 0.556 | 0.478 | 0.520 | 0.532 | 0.580 |
| EFGE-Pois | 0.460 | 0.504 | 0.501 | 0.518 | 0.497 | 0.490 | 0.491 | 0.562 | 0.497 | 0.491 | 0.499 | 0.566 |
| EFGE-Norm | 0.456 | 0.496 | 0.503 | 0.526 | 0.473 | 0.500 | 0.516 | 0.533 | 0.471 | 0.505 | 0.533 | 0.581 |
| Splitter | 0.164 | 0.162 | 0.183 | 0.181 | 0.169 | 0.166 | 0.188 | 0.186 | 0.165 | 0.162 | 0.166 | 0.177 |
| Dp-Bern | 0.461 | 0.481 | 0.528 | **0.589** | 0.478 | 0.519 | 0.533 | 0.563 | 0.504 | 0.540 | 0.564 | 0.559 |
| Dp-Pois | 0.435 | 0.462 | 0.479 | 0.538 | 0.430 | 0.460 | 0.476 | 0.528 | 0.403 | 0.441 | 0.460 | 0.510 |
| Dp-Norm | 0.475 | 0.490 | 0.510 | 0.556 | 0.509 | 0.523 | 0.529 | **0.559** | 0.512 | 0.529 | 0.527 | 0.562 |
| up-Bern | 0.459 | 0.492 | 0.509 | 0.538 | 0.481 | 0.522 | **0.534** | 0.538 | 0.502 | 0.557 | 0.581 | 0.585 |
| up-Pois | 0.437 | 0.465 | 0.498 | 0.540 | 0.436 | 0.467 | 0.492 | 0.546 | 0.404 | 0.438 | 0.473 | 0.529 |
| up-Norm | **0.518** | **0.527** | **0.532** | 0.568 | **0.521** | **0.531** | 0.514 | 0.561 | **0.521** | **0.559** | **0.580** | **0.616** |

| Yeast | (D = 50) | | | | (D = 100) | | | | (D = 150) | | | |
|---|---|---|---|---|---|---|---|---|---|---|---|---|
| Deepwalk | 0.283 | 0.330 | 0.360 | 0.413 | 0.290 | 0.358 | 0.401 | 0.436 | 0.288 | 0.361 | 0.400 | 0.441 |
| node2vec | 0.280 | 0.320 | 0.351 | 0.388 | 0.293 | 0.338 | 0.371 | 0.410 | 0.297 | 0.354 | 0.401 | 0.437 |
| struc2vec | 0.134 | 0.150 | 0.169 | 0.256 | 0.134 | 0.153 | 0.166 | 0.238 | 0.141 | 0.161 | 0.171 | 0.225 |
| EFGE-Bern | 0.269 | 0.324 | 0.347 | 0.380 | 0.281 | 0.339 | 0.374 | 0.418 | 0.289 | 0.349 | 0.400 | 0.414 |
| EFGE-Pois | 0.271 | 0.320 | 0.365 | 0.373 | 0.281 | 0.331 | 0.372 | 0.399 | 0.286 | 0.339 | 0.374 | 0.409 |
| EFGE-Norm | 0.285 | 0.325 | 0.354 | 0.383 | 0.281 | 0.332 | 0.367 | 0.405 | 0.288 | 0.354 | 0.392 | 0.428 |
| Splitter | 0.164 | 0.207 | 0.228 | 0.246 | 0.157 | 0.214 | 0.263 | 0.263 | 0.165 | 0.211 | 0.273 | 0.297 |
| Dp-Bern | 0.285 | **0.343** | 0.373 | 0.401 | 0.296 | 0.377 | 0.402 | 0.442 | 0.296 | **0.376** | 0.416 | 0.472 |
| Dp-Pois | 0.275 | 0.328 | 0.354 | 0.375 | 0.285 | 0.327 | 0.360 | 0.383 | 0.301 | 0.338 | 0.375 | 0.402 |
| Dp-Norm | 0.285 | 0.330 | 0.364 | 0.352 | 0.277 | 0.339 | 0.352 | 0.381 | 0.266 | 0.350 | 0.382 | 0.407 |
| up-Bern | **0.290** | 0.338 | **0.382** | **0.414** | **0.301** | 0.361 | **0.406** | **0.443** | **0.304** | 0.367 | **0.419** | **0.479** |
| up-Pois | 0.281 | 0.336 | 0.358 | 0.392 | 0.288 | 0.326 | 0.355 | 0.385 | 0.277 | 0.348 | 0.395 | 0.426 |
| up-Norm | 0.282 | 0.340 | 0.372 | 0.393 | 0.289 | 0.345 | 0.391 | 0.381 | 0.288 | 0.320 | 0.364 | 0.382 |

*AAAI Conference on Artificial Intelligence*, volume 34, pages 3357–3364, 2020.

Yujun Chen, Juhua Pu, Xingwu Liu, and Xiangliang Zhang. Gaussian mixture embedding of multiple node roles in networks. *World Wide Web*, 23(2):927–950, 2020.

Gabor Csardi and Tamas Nepusz. The igraph software package for complex network research. *InterJournal*, Complex Systems:1695, 2006. URL https://igraph.org.

Alessandro Epasto and Bryan Perozzi. Is a single embedding enough? learning node representations that capture multiple social contexts. In *The world wide web conference*, pages 394–404, 2019.

Aditya Grover and Jure Leskovec. node2vec: Scalable feature learning for networks. In *Proceedings of the 22nd ACM SIGKDD international conference on Knowledge discovery and data mining*, pages 855–864, 2016.

Viet Huynh, Dinh Phung, and Svetha Venkatesh. Streaming variational inference for dirichlet process mixtures. In *Asian Conference on Machine Learning*, pages 237–252. PMLR, 2016.

Diederik P Kingma and Jimmy Ba. Adam: A method for stochastic optimization. In *ICLR (Poster)*, 2015.

Qimai Li, Zhichao Han, and Xiao-Ming Wu. Deeper insights into graph convolutional networks for semi-supervised learning. In *Thirty-Second AAAI conference on artificial intelligence*, 2018.

Ninghao Liu, Qiaoyu Tan, Yuening Li, Hongxia Yang, Jingren Zhou, and Xia Hu. Is a single vector enough? exploring node polysemy for network embedding. In *Proceedings of the 25th ACM SIGKDD International Conference on Knowledge Discovery & Data Mining*, pages 932–940, 2019.

Tomas Mikolov, Kai Chen, Greg Corrado, and Jeffrey Dean. Efficient estimation of word representations in vector space. *arXiv preprint arXiv:1301.3781*, 2013a.

Tomas Mikolov, Ilya Sutskever, Kai Chen, Greg S Corrado, and Jeff Dean. Distributed representations of words and phrases and their compositionality. In *Advances in neural information processing systems*, pages 3111–3119, 2013b.

Mingdong Ou, Peng Cui, Jian Pei, Ziwei Zhang, and Wenwu Zhu. Asymmetric transitivity preserving graph embedding. In *Proceedings of the 22nd ACM SIGKDD international conference on Knowledge discovery and data mining*, pages 1105–1114, 2016.

Chanyoung Park, Carl Yang, Qi Zhu, Donghyun Kim, Hwanjo Yu, and Jiawei Han. Unsupervised differentiable multi-aspect network embedding. In *Proceedings of the*

*26th ACM SIGKDD International Conference on Knowledge Discovery & Data Mining*, pages 1435–1445, 2020.

Bryan Perozzi, Rami Al-Rfou, and Steven Skiena. Deepwalk: Online learning of social representations. In *Proceedings of the 20th ACM SIGKDD international conference on Knowledge discovery and data mining*, pages 701–710, 2014.

Jiezhong Qiu, Yuxiao Dong, Hao Ma, Jian Li, Kuansan Wang, and Jie Tang. Network embedding as matrix factorization: Unifying deepwalk, line, pte, and node2vec. In *Proceedings of the eleventh ACM international conference on web search and data mining*, pages 459–467, 2018.

Leonardo FR Ribeiro, Pedro HP Saverese, and Daniel R Figueiredo. struc2vec: Learning node representations from structural identity. In *Proceedings of the 23rd ACM SIGKDD international conference on knowledge discovery and data mining*, pages 385–394, 2017.

Ryan A. Rossi and Nesreen K. Ahmed. The network data repository with interactive graph analytics and visualization. In *AAAI*, 2015. URL https://networkrepository.com.

Benedek Rozemberczki, Oliver Kiss, and Rik Sarkar. Karate Club: An API Oriented Open-source Python Framework for Unsupervised Learning on Graphs. In *Proceedings of the 29th ACM International Conference on Information and Knowledge Management (CIKM '20)*, page 3125–3132. ACM, 2020.

Maja Rudolph and David Blei. Dynamic embeddings for language evolution. In *Proceedings of the 2018 World Wide Web Conference*, pages 1003–1011, 2018.

Maja Rudolph, Francisco Ruiz, Stephan Mandt, and David Blei. Exponential family embeddings. In *Advances in Neural Information Processing Systems*, pages 478–486, 2016.

Maja Rudolph, Francisco Ruiz, Susan Athey, and David Blei. Structured embedding models for grouped data. *Advances in neural information processing systems*, 30, 2017.

Jayaram Sethuraman. A constructive definition of dirichlet priors. *Statistica sinica*, pages 639–650, 1994.

Fan-Yun Sun, Meng Qu, Jordan Hoffmann, Chin-Wei Huang, and Jian Tang. vgraph: A generative model for joint community detection and node representation learning. *Advances in Neural Information Processing Systems*, 32, 2019.

Petar Velickovic, William Fedus, William L Hamilton, Pietro Liò, Yoshua Bengio, and R Devon Hjelm. Deep graph infomax. *ICLR (Poster)*, 2(3):4, 2019.

Hanna Wallach, Shane Jensen, Lee Dicker, and Katherine Heller. An alternative prior process for nonparametric bayesian clustering. In *Proceedings of the Thirteenth International Conference on Artificial Intelligence and Statistics*, pages 892–899. JMLR Workshop and Conference Proceedings, 2010.

Xiao Wang, Peng Cui, Jing Wang, Jian Pei, Wenwu Zhu, and Shiqiang Yang. Community preserving network embedding. In *Thirty-first AAAI conference on artificial intelligence*, 2017.

Zonghan Wu, Shirui Pan, Fengwen Chen, Guodong Long, Chengqi Zhang, and S Yu Philip. A comprehensive survey on graph neural networks. *IEEE transactions on neural networks and learning systems*, 32(1):4–24, 2020.
