# OpenReview forum: "Nonparametric Exponential Family Graph Embeddings for Multiple Representation Learning"
_auai.org/UAI/2022/Conference — UAI 2022 Poster_

### Official Review · Reviewer_TZYv · 2022-03-28

**Q2(1) Originality/Novelty:** 2
**Q2(2) Significance/Impact:** 2
**Q2(3) Correctness/Technical Quality:** 3
**Q2(6) Clarity Of Writing:** 2
**Q6 Overall Score:** 5
**Q8 Confidence In Your Score:** 3

**Q1 Summary And Contributions:**

This paper summarized the graph embedding framework under the bayesian consideration. Specially, they discussed different prior distributions.

**Q2 Assessment Of The Paper:**

More detailed information regarding each of these aspects is given below:

**Q2(4) Quality Of Experiments (Optional):**

2: Fair: The experimental evaluation is weak: important baselines are missing, or the results do not adequately support the main claims.

**Q2(5) Reproducibility:**

2: Fair: Key resources (e.g., proofs, code, data) are unavailable but key details (e.g., proof sketches, experimental setup) are sufficiently well-described for an expert to confidently reproduce the main results.

**Q3 Main Strengths:**

It is a very nice discussion to illustrate different distributions prior to node embedding.
Judged from the results, we do see the variance in performance by different considerations.


**Q4 Main Weakness:**

However, simply applying different prior distributions could not be considered as innovative in current graph learning fields.
I also have a hard time understanding what is "multiple" representation learning? Judging from the paper, it is a different level of dimensions of the parameter?
Since you introduce the context vector and embedding vector. Will they contribute differently? How important is each vector to the overall embedding?
There are some great variations in the performance of the proposed model.  Like dp-emb pos and up-emb pos have worse performance. How can you choose the best model?

**Q5 Detailed Comments To The Authors:**

Please see above.

**Q7 Justification For Your Score:**

This paper has certain merit in summarizing the probabilistic embedding of graphs.
However, it does not necessarily generate new thoughts in the embedding field.

**Q9 Complying With Reviewing Instructions:**

1: Yes.

---

### Official Review · Reviewer_DWy3 · 2022-04-11

**Q2(1) Originality/Novelty:** 2
**Q2(2) Significance/Impact:** 1
**Q2(3) Correctness/Technical Quality:** 2
**Q2(6) Clarity Of Writing:** 2
**Q6 Overall Score:** 4
**Q8 Confidence In Your Score:** 3

**Q1 Summary And Contributions:**

The paper proposes modifying exponential family embeddings (EFE) to use a nonparametric prior over each vertices embedding vector to encourage more flexible representations.
A variational inference scheme is devised to perform approximate inference in this model.
The paper evaluates the model's embedding vectors in link prediction and node classification tasks.

**Q2 Assessment Of The Paper:**

More detailed information regarding each of these aspects is given below:

**Q2(4) Quality Of Experiments (Optional):**

2: Fair: The experimental evaluation is weak: important baselines are missing, or the results do not adequately support the main claims.

**Q2(5) Reproducibility:**

3: Good: Key resources (e.g., proofs, code, data) are available and key details (e.g., proofs, experimental setup) are sufficiently well-described for competent researchers to confidently reproduce the main results.

**Q3 Main Strengths:**

The paper extends EFE to allow for a nonparametric prior (either Dirichlet or Uniform Process) over each vertices' embedding vector $\rho_v^{(n)}$. This is an interesting idea to allow for each vertex to weight its context $\tilde{x}_{c_n}$ differently.

**Q4 Main Weakness:**

The paper does not highlight the practical significant advantages of the extension or the technical challenges the paper overcomes in developing it.

The presentation is confusing. Specifically for the "generative process", the definition of observed value $x_{n,v}$ in the beginning of Section 3.1 describes it as the one-hot encoding of a random walk (plus negative sampling later), but Equation (4) and the generative process at the end of Section 3 suggests it comes from an exponential family.

The experiments are weak. Why and how does the multiple embedding vectors for each vector help with link prediction / classification. The logistic regression details are missing?

**Q5 Detailed Comments To The Authors:**

The paper should be revised to highlight the *novel*/*significant* contributions of extending EFE to include a nonparametric prior. There is a lot of very good technical work here, but the advantage/need for "multiple latent embedding vectors" for each vertex is not conveyed by the experiments; it looks like the logistic regression just uses the $\beta$ weighted mean embedding (?)). Similarly, the model / inference in Section 3 / 4 don't highlight novel challenges overcome by this work.

For the "generative process": I believe the idea is that we generate observations from random walks on the graph (and negative sampling), and then fit an embedding model to these random walk observations to obtain embedding vectors for each vertex. Correct?

The presentation could be improved to clarify that a location "n" is dependent on the current random walk in consideration (as we later consider observations from multiple random walks).

Similarly, it would help if the "context" of location "n" is clarified to be stochastic / dependent on the random walk in consideration (to contrast with a fixed "neighborhood" of the vertex).

**Q7 Justification For Your Score:**

The paper appears to be an incremental improvement of EFE. The paper does not highlight what the practical/empirical motivation for a nonparametric prior nor any novel contributions to the dataset generation / inference scheme.


**Q9 Complying With Reviewing Instructions:**

1: Yes.

---

### Official Review · Reviewer_XJcM · 2022-04-14

**Q2(1) Originality/Novelty:** 2
**Q2(2) Significance/Impact:** 2
**Q2(3) Correctness/Technical Quality:** 2
**Q2(6) Clarity Of Writing:** 3
**Q6 Overall Score:** 6
**Q8 Confidence In Your Score:** 3

**Q1 Summary And Contributions:**

The paper focuses on graph embedding methods, with the goal of proposing a solution which is able to provide multiple latent vector representations for a node, in order to take into account additional node information (e.g. being hug or bridge) besides the co-appearance pattern of nodes in walks across a graph. An experimental evaluation is provided on node classification and link prediction tasks, showing improved performances with respect to the state of the art.

**Q10 Ethical Concerns (Optional):**

No ethical concerns can be found

**Q2 Assessment Of The Paper:**

More detailed information regarding each of these aspects is given below:

**Q2(4) Quality Of Experiments (Optional):**

2: Fair: The experimental evaluation is weak: important baselines are missing, or the results do not adequately support the main claims.

**Q2(5) Reproducibility:**

3: Good: Key resources (e.g., proofs, code, data) are available and key details (e.g., proofs, experimental setup) are sufficiently well-described for competent researchers to confidently reproduce the main results.

**Q3 Main Strengths:**

- The paper is overall clear and well written
- The experimental evaluation shows improvements (sometimes moderate) over state of the art solutions
- The study of the exploitation of the normal distribution in the proposed solution would be useful for the community

**Q4 Main Weakness:**

- Moderate novelty

- The extension to multiple context vectors for a vertex is somehow urgent, since this is one of the key aspects highlighted by the authors.

- The discussion of the experimental results is rather limited, it should be definitely substantiated.

**Q5 Detailed Comments To The Authors:**

The paper is overall clear and well written, enriched by a brief but effective analysis of related works.

The proposed solution extends the Exponential family graph embedding model with two nonparametric prior settings: the Dirichlet process and the uniform process. The model combines the ability of Exponential family graph embedding to take the number of occurrences of context nodes into account with nonparametric priors giving it the flexibility to learn more than one latent representation for each node.

Overall the proposed method shows limited novelty since it mostly adopts and combines existing solutions. However the experimental evaluation shows improvements (sometimes moderate) over state of the art solutions. The most interesting part is the study of the exploitation of the normal distribution in the proposed solution.

The description of the experimental evaluation resulted overall appropriate. Datasets and source code should be publicly available and not only provided as supplemental material.

The are some aspects of the paper that could be refined. Details are provided in the following.
- In the introduction, the authors should specify what is meant for different functions/roles of nodes.
- Figure 1 is useful but, besides the caption, it would be preferable to have additional text describing it.
- The discussion of the experimental results is rather limited, it should be definitely substantiated.
- The extension to multiple context vectors for a vertex is somehow urgent, since this is one of the key aspects highlighted by the authors.

**Q7 Justification For Your Score:**

The paper shows moderate novelty but rather interesting experimental results. The study of the exploitation of the normal distribution in the proposed solution would be useful for the community. There is space for improving the paper.

**Q9 Complying With Reviewing Instructions:**

1: Yes.

---

### Official Review · Reviewer_mRGL · 2022-04-16

**Q2(1) Originality/Novelty:** 3
**Q2(2) Significance/Impact:** 3
**Q2(3) Correctness/Technical Quality:** 3
**Q2(6) Clarity Of Writing:** 3
**Q6 Overall Score:** 7
**Q8 Confidence In Your Score:** 2

**Q1 Summary And Contributions:**

The authors put forth a nonparametric exponential family graph embedding, with multiple functionalities. They empirically demonstrate the learned multiple representations can enhance performance in two tasks (link prediction and node classification).


**Q2 Assessment Of The Paper:**

More detailed information regarding each of these aspects is given below:

**Q2(4) Quality Of Experiments (Optional):**

3: Good: The experimental evaluation is adequate, and the results convincingly support the main claims.

**Q2(5) Reproducibility:**

3: Good: Key resources (e.g., proofs, code, data) are available and key details (e.g., proofs, experimental setup) are sufficiently well-described for competent researchers to confidently reproduce the main results.

**Q3 Main Strengths:**

- Intuitive and principled extension of exponential family graph embedding to non-parameteric setting
- Inference of embedding vectors algorithms for three different exponential family distributions: Bernoulli, Poisson, and Gaussian
- Considered two possible process prior: Dirichlet and Uniform
- Experiments indicate their model outperforms previously known models


**Q4 Main Weakness:**

- Experiments could have been more extensive


**Q5 Detailed Comments To The Authors:**

- Would it be interesting to study the exponential family in the geometric and chi squared distributions?
- Please consider a more elaborate exposition of Section 2.1, much like Section 2 of Rudolph et al. This would help the reader not familiar with the topic.


**Q7 Justification For Your Score:**

I’m not very familiar with this area of research, but from what I understood, the paper seems above the bar for UAI.


**Q9 Complying With Reviewing Instructions:**

1: Yes.

---

### Decision · Program_Chairs · 2022-05-15

**Decision:**

Accept (Poster)

**Comment:**

Meta Review: This is an interesting and well-written paper. Perhaps it it less obvious however, how it contributes to the existing literature. It clearly builds heavily on earlier work and only the rebuttal made it a bit clear (at least to me) what is innovative about the proposed approach. There was a consensus among the reviewers that the paper could be accepted if there is space. I ask the authors to incorporate the clarifying comments they sent to the referees - I think they are really crucial to improve the paper. I also realized that none of the referees pointed out that the "exponential family embeddings" are really just special generalized linear models. I think it is important to keep track of the original ideas and it would be good if the authors should mention generalized linear models explicitly (as the paper proposing "exponential family embeddings" does).